# Current Trends and Applications of Machine Learning in Tribology—A Review

**Max Marian** [1,*] and **Stephan Tremmel** [2]

1 Engineering Design, Friedrich-Alexander-University Erlangen-Nuremberg (FAU), Martensstr. 9, 91058 Erlangen, Germany
2 Engineering Design and CAD, University of Bayreuth, Universitätsstr. 30, 95477 Bayreuth, Germany; stephan.tremmel@uni-bayreuth.de
* Correspondence: marian@mfk.fau.de

**Abstract:** Machine learning (ML) and artificial intelligence (AI) are rising stars in many scientific disciplines and industries, and high hopes are being pinned upon them. Likewise, ML and AI approaches have also found their way into tribology, where they can support sorting through the complexity of patterns and identifying trends within the multiple interacting features and processes. Published research extends across many fields of tribology from composite materials and drive technology to manufacturing, surface engineering, and lubricants. Accordingly, the intended usages and numerical algorithms are manifold, ranging from artificial neural networks (ANN), decision trees over random forest and rule-based learners to support vector machines. Therefore, this review is aimed to introduce and discuss the current trends and applications of ML and AI in tribology. Thus, researchers and R&D engineers shall be inspired and supported in the identification and selection of suitable and promising ML approaches and strategies.

**Keywords:** tribology; machine learning; artificial intelligence; triboinformatics; databases; data mining; meta-modeling; artificial neural networks; monitoring; analysis; prediction; optimization

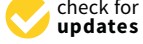



## 1. Introduction

Tribology has been and continuous to be one of the most relevant fields in today's society, being present in almost aspects of our lives. The importance of friction, lubrication and wear is also reflected by the significant share of today's world energy consumption [1]. The understanding of tribology can pave the way for novel solutions for future technical challenges. At the root of all advances are multitudes of precise experiments and advanced computer simulations across different scales and multiple physical disciplines [2]. In the context of tribology 4.0 [3] or triboinformatics [4], advanced data handling, analysis, and learning methods can be developed based upon this sound and data-rich foundation and employed to expand existing knowledge. Moreover, tribology is characterized by the fact that it is not yet possible to fully describe underlying processes with mathematical terms, e.g., by differential equations. Therefore, modern Machine Learning (ML) or Artificial Intelligence (AI) methods provide opportunities to explore the complex processes in tribological systems and to classify or quantify their behavior in an efficient or even real-time way [5]. Thus, their potential also goes beyond purely academic aspects into actual industrial applications. The advantages and the potential of ML and AI techniques are seen especially in their ability to handle high dimensional problems and data sets as well as to adapt to changing conditions with reasonable effort and cost [6]. They allow for the identification of relevant relations and/or causality, thus expand the existing knowledge with already available data. Ultimately, through analyses, predictions, and optimizations, transparent and precise recommendations for action could be derived for the engineer, practitioner, or even the potentially smart and adaptive tribological system itself. Nevertheless, compared

to other disciplines or domains, e. g. economics and finances [7], health care [8], or manufacturing processes [6], the applicability of ML and AI techniques for tribological issues is still surprisingly underexplored. This is certainly also due to the interdisciplinarity and the quantity of heterogenous data from simulations on different scales or manyfold measurement devices with individual uncertainties. Furthermore, friction and wear characteristics do not represent hard data, but irreversible loss quantities with a dependence on time and test conditions [9].

To help pave the way, a more detailed analysis of the available ML/AI techniques as well as their applicability, strengths and limitations with regard to the requirements of the respective tribological application scenario with its specific, theoretical foundations is essential. Therefore, this contribution aims to introduce the trends and applications of ML algorithms with relevance to the domain of tribology. While other reviews were more generic [10], had a more concise scope [5], or focused on a specific technique (i.e., artificial neural networks [11]), this review article is also intended to cover a wider range of techniques and in particular to shed light on the broad applicability to various fields with tribological issues. Thus, the interested reader shall be provided with a high-level understanding of the capabilities of certain methods with respect to the tribological applications ranging from composite materials over drive technology or manufacturing to surface engineering and lubricant formulations. This article is therefore structured in such a way that first the theoretical background is introduced, and the results of a quantitative meta-literature analysis are presented (Section 2). Thereby, the published work on ML in the field of tribology is clustered according to the level or intention, the scale under consideration, the nature of the database and the area of application. Organized according to the latter, the work and progress reported in literature is then discussed in detail (Sections 3.1–3.6) before the main trends are summarized and concluded (Section 4).

## 2. Background and a Quantitative Survey on Machine Learning in Tribology

ML is part of AI [12] and thus originally a sub-domain of computer science. AI and ML are formed by logic, probability theory, algorithm theory, and computing [13]. In a first step, ML involves designing computing systems for a special task that can learn from training data over time and develop and refine experience-based models that predict outcomes. The system can thus be used to answer questions in the given field [12]. There are a number of different algorithms that can be used for ML, whereby the suitability is strongly task-dependent. Generally, algorithms can be categorized as "supervised learning" or "unsupervised learning" [12]. For the former, algorithms learn a relation from a given set of input and output data vectors. During learning, a "teacher" (e.g., an expert) provides the correct inputs and outputs. In unsupervised learning, the algorithm generates a statistical model that describes a given data set without the model being evaluated by a "teacher". Furthermore, reinforcement learning features different characteristics, although it is sometimes classified as supervised learning. Instead of induction from pre-classified examples, an "agent" "experiments" with the system and the system responds to the experiments with reward or punishment. The agent thus optimizes the behavior with the goal of maximizing reward and minimizing punishment. While the classification of the three learning types mentioned above is common and widely accepted, there is no consensus on which algorithms should be assigned to which category. One possible allocation following [6] is illustrated in Figure 1.

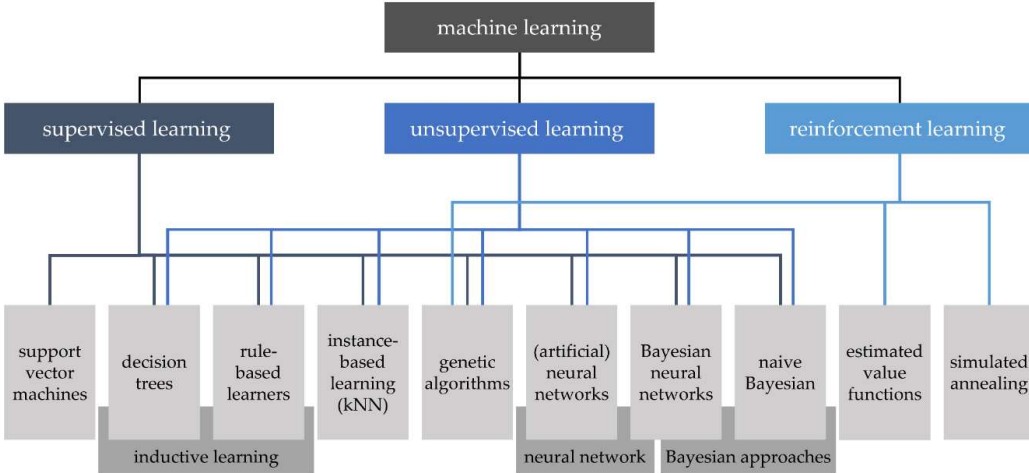

**Figure 1.** Classification of machine learning techniques. Redrawn from [6] with permission by CC BY 4.0 (Taylor and Francis).

The basic idea of *support vector machines* (SVM) is that a known set of objects is represented by a vector in a vector space. Hyperplanes are introduced into this space to separate the data points. In most cases, only the subset of the training data that lies on the boundaries of two planes is relevant. These vectors are the name-giving support vectors [14]. To account for nonlinear boundaries, kernel functions are an essential part of SVM. By using the kernel trick, the vector space is transformed into an arbitrarily higher-dimensional space, so that arbitrarily nested vector sets are linearly separable [15]. *Decision trees* (DT) are ordered, directed trees that illustrate hierarchically successive decisions [12]. A decision tree always consists of a root node and any number of inner nodes as well as at least two leaves. Each node represents a logical rule, and each leaf represents an answer to the decision problem. The complexity and semantics of the rules are not restricted, although all decision trees can be reduced to binary decision trees. In this case, each rule expression can take only one of two values [16]. A possibility to increase the classification quality of decision trees is the use of sets of decision trees instead of single trees, this is called *decision forests* [17]. If decision trees are uncorrelated, they are called *random forest* (RF) [18]. The idea behind decision forests is that while a single, weak decision tree may not provide optimal classification, a large number of such decision trees are able to do so. A widely used method for generating decision forests is boosting [19]. In *rule-based learners*, the output results from composing individual rules, which are typically expressed in the form "If–Then". Rule-based ML methods typically comprise a set of rules, that collectively make up the prediction model. *K-Nearest-Neighbor algorithms* (kNN) are classification methods in which class assignment is performed considering *k* nearest neighbors, which were classified before. The choice of *k* is crucial for the quality of the classification [16]. In addition, different distance measures can be considered [20]. *Artificial neural networks* (ANN) are essentially modeled on the architecture of natural brains [21]. They are 'a computing system made up of a number of quite simple but highly interconnected processing elements (neurons), which process information by their dynamic state response to external inputs' [12]. The so-called transfer function calculates the neuron's network input based on the weighting of the inputs [22]. Calculating the output value is done by the so-called activation function considering a threshold value [12,22]. Weightings and thresholds for each neuron can be modified in a training process [16]. The overall structure of neurons and interconnections, in particular how many neurons are arranged in a layer and how many neurons are arranged in parallel per layer, is called topology or architecture. The last layer is called the output layer and there can be several hidden layers between the input and the output layer (multilayer ANN) [21]. While single-layer networks can only be used to solve linear problems, multi-layer networks also allow the solution of nonlinear problems [12]. Feedforward means, that neuron outputs are routed in processing direction only. Recurrent networks, in contrast,

also have feedback loops. Commonly, ANNs are represented in graph theory notation, with nodes representing neurons and edges representing their interconnections.

Already rather early works in the field of tribology from the 1980s can be assigned to the current understanding of ML. For example, Tallian [23,24] introduced computerized databases and expert systems to support tribological design decisions or failure diagnosis. Other initial studies were concerned, for example, with the prediction of tribological properties [25–27] or classification of wear particles [28,29]. Between 1985 and today, almost 130 publications related to ML in tribology were identified within a systematic literature review (see Prisma flow chart in Figure 2a), whereas the number of papers initially increased slowly and more rapidly within the last decade (Figure 2b). During the latter period, the number of publications has more than tripled, which represents a faster growth than the general increase in the number of publications in the field of tribology (the numbers of Scopus-listed publications related to tribology grew by a factor of 2.3 between 2010 and today). It can therefore be highly expected that this trend will continue and that ML techniques will also become increasingly prominent in the field of tribology due to technological advances and decreasing barriers and preconceptions. Therefore, the analysis of the publications with respect to the fields of application is of particular interest, which is illustrated in Figure 2c. Especially in the areas of composite materials, drive technology, and manufacturing, numerous successful implementations of ML algorithms can already be found. Yet, some studies can also be found for surface engineering, lubricant formulation or manufacturing. As depicted in Figure 2d, ML techniques are applied for monitoring tribo-systems or for pure analytical/diagnostic purposes, but especially for predicting and optimizing the tribological behavior with respect to the friction and wear behavior. The scales under consideration are mainly on the macro and/or micro level, see Figure 2e. However, a few works also show the applicability down to the nano scale. Finally, it could be observed that the database for training the ML algorithms can also be generated based on numerical or theoretical fundamentals from simulation models or on information from the literature. However, the vast majority (roughly three quarters) of the published work is based upon experimentally generated data sets (Figure 2f).

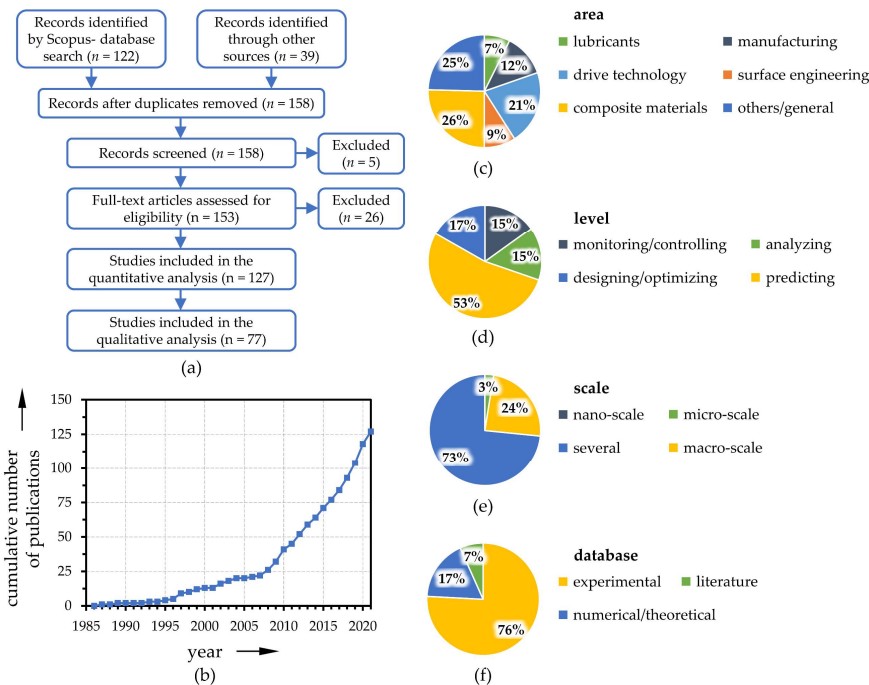

**Figure 2.** Systematic protocol (Prisma flow chart) for the paper collection/screening (**a**), and number of publications per year (**b**) and clustered by the area of application (**c**), level (**d**), scale (**e**) and database (**f**).

## 3. Results

As illustrated in the previous section, there is a wide variety of implementations of different ML/AI approaches. In order to give a more detailed overview of applications for ML to solve tribological issues, various cases are presented and discussed in the following. Since the aim is to address interested readers from the field of tribology and to show how ML can be used effectively in their respective fields, this is organized according to the area of application in descending order of the number of published works.

### 3.1. Composite Materials

ML and AI algorithms are already widely used in the field of composite materials for tribological applications. Generally, there has been a remarkable growth in the large-scale use of materials made from two or more constituent materials with different physical or chemical properties, for example a fiber and/or filler reinforced polymer (PMC), ceramic (CMC) or metal (MMC) matrix composites. The advantages of these materials lie especially in the high strength-to-weight as well as stiffness-to-weight ratios [30]. For a general overview of tribological properties of different composites in dependency of contact and/or environmental conditions, the interested reader is referred to various review articles [31–33]. A major field which already exploited ML approaches to a greater extent have been wear-resistant composites with polymer matrix, for example thermosets such as epoxy or polyester [34] as well as thermoplastics [35], e.g., polyamide (PA), polyphenylene sulfide (PPS), polytetrafluoroethylene (PTFE), polyethylene (PE), polyether ether ketone (PEEK) [36,37], or polypropylene (PP) [38].

### 3.1.1. Thermoset Matrix Composites

In this way, Padhi and Satapathy [39] applied a Taguchi experimental design of experiments (DoE, 16 data points) in combination with a back propagation ANN to train multi-layered feed-forward networks, predicting the tribological behavior of epoxy composites with short glass fibers (SGF) and/or micro-sized blast furnaces slag (BFS) particles. Based on data obtained from tests in a pin-on-disk setup under dry sliding conditions against a hardened ground steel counter-body and divided into training, test and validation categories and operational and material parameters with significance for the resulting wear rate were thus identified. Thereby, the ANN was able to predict the specific wear rate with low errors between 2.5% and 6.9% for composites without BFS and between 0.9% and 5.1% for composites with BFS. Epoxy composites were also investigated recently by Egala et al. [40] with newly developed natural short castor oil fibers (ricinus communis) as unidirectional reinforcements of different lengths and at a constant volume fraction of 40%. The database consisting of 36 data points was acquired from experiments utilizing a flat pin-on-disk tribometer under dry sliding conditions against a hardened steel disk as a counter-body. Besides fiber lengths, the normal force as well as the sliding distance were varied and the influence on gravimetric wear, interfacial heat, and COF were studied within a full factorial DoE. The experiments were carried out in duplicate and averaged values were used in further data processing. Thereby, the relationships between variation parameters and target values were expressed by linear regression as well as by hidden layer ANNs. For the training of the latter, the data set was randomly split into training (60%), validation (20%), and test (20%) data. To find the best prediction, 73 different ANNs (cascade forward back propagation, feed forward back propagation and layer recurrent) with Levenberg-Marquardt (LM) training function and a varying number of hidden layers (1–4), number of neurons (7–15), and different transfer functions (Logsig, Purelin) were tested stepwise (see Figure 3a–d). It was found that the linear regressions were able to describe the results within errors of ±8%. The best predictions however were provided by a cascade forward back propagation network as well as a feed forward back propagation ANN with architectures as illustrated in Figure 3e,f using Trainlm and Purelin as training and transfer functions. Thereby, the errors were ±5% and ±4.5%, respectively,

indicating higher efficiency and reliability in predicting the tribological behavior of studied composites than common regression models.

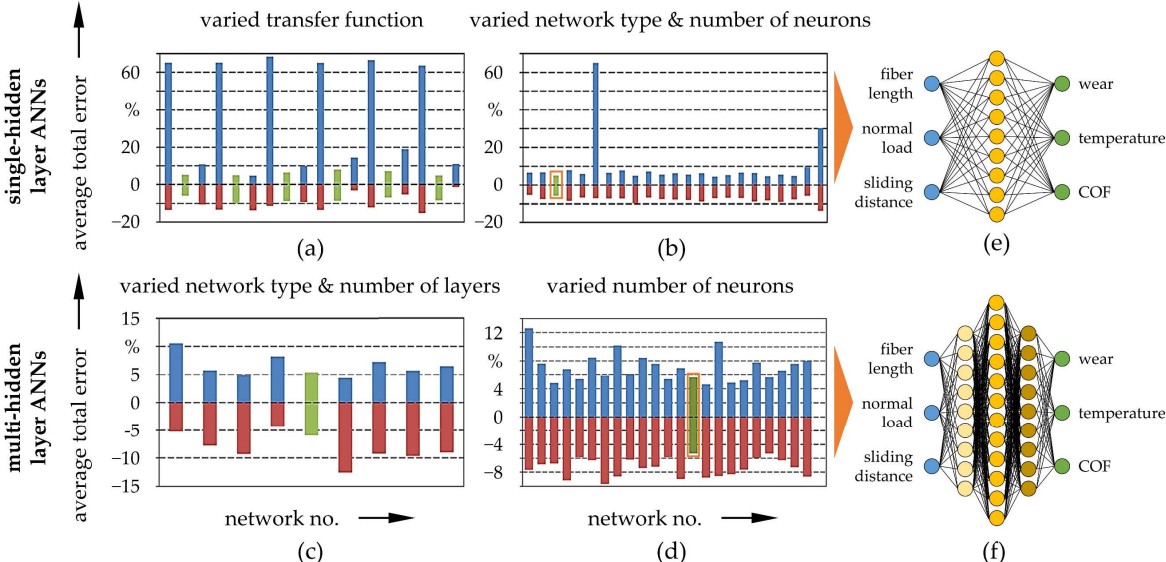

**Figure 3.** Average total errors for the wear prediction of different ANN architectures when optimizing the transfer function (**a**), the network type and the number of neurons in the single-hidden layer ANN (**b**) as well as the network type and the number of layers (**c**) and the number of neurons in the multi-hidden layer ANN (**d**). Illustration of the architectures of the single- (**e**) and multi-hidden layer ANN (**f**) with least errors. Redrawn from [40] with permission by CC BY 4.0 (Springer).

Nirmal [41] attempted to predict the friction coefficient of treated betelnut fiber reinforced polyester composites by an ANN trained with data from 492 experimental sets of a block-on-disk tribometer against stainless steel under dry sliding conditions with varying normal loads, sliding distances and three different fiber orientations (parallel, anti-parallel and normal). In trial-and-error variations of neuron, layer, and transfer function, an ANN consisting of two hidden layers with 30 and 20 neurons, respectively, trained by LM function and utilizing logsig transfer functions between the hidden layers and a pure linear transfer function to the output layer was found as most capable of predicting the COF based upon the inputs. Albeit other training algorithms (gradient descent back propagation, with momentum and adaptive learning rate, with adaptive learning rate and conjugate gradient back propagation with Powell-Beale restarts) resulted in significantly faster convergence, the LM function featured the lowest errors compared to the test data, especially after repeated training. Thus, sum squared errors (SSE) of less than $10^{-2}$ were obtained. Similarly, Nasir et al. [42] identified the LM function as most suitable compared to others when training ANNs to predict the COF from 7389 data sets attained in experiments on multi-layered glass fiber reinforced polyester resin rubbing against stainless steel using a disk-on-flat tribometer under different fiber orientations, loads, sliding speeds, and test durations. The prediction model was able to reproduce the trends of the experiments well and accuracies up to 90% were achieved. It was stated, however, that performance was lower compared to other studies due to the large amount of input data as well as larger deviations and fluctuations in the experimental results, especially during running-in periods. Furthermore, it was emphasized that the number of layers as well as neurons have a decisive influence on the results. While multi-hidden layer ANNs mapped partial areas of the input data (e.g., only one fiber orientation) very well, the entire data area was best represented by a single-hidden layer ANN with comparatively many neurons.

### 3.1.2. Thermoplastic Matrix Composites

Already in the early 2000s, Velten et al. [43,44] evaluated the ability of ANNs to predict tribological properties of short fiber thermoplastic matrix (PA) composites and aid in the

material design. Here, the decisive role of the data sets as well as the ANN architecture was emphasized as well. Later, Gyurova et al. [45] modeled the tribological behavior of PPS composites with short carbon fibers (SCF), graphite, PTFE, and titanium dioxide (TiO$_2$) fillers with over 90 data sets obtained from dry-running pin-on-disk tribometer tests at constant test duration and varied loads and sliding speeds. The data were split into 80% training and 20% testing data and included the material composition (matrix volume fraction, filler, reinforcing agents and lubricants), testing conditions (pressure and sliding speed), as well as characteristic thermo-mechanical properties (tensile and compressive properties) as inputs and the specific wear and the friction coefficient as outputs. For the latter, separate ANNs were trained by a gradient descent back propagation algorithm with momentum and adaptive learning rate to minimize the mean relative error (MRE). These consisted of two hidden layers with 9 and 3 (wear rate) as well as 3 and 1 (COF) neurons, respectively. Thus, most significant inputs could be identified, and it was observed that the MRE for the wear rate (0.60–0.78) was higher than for the sliding friction (0.10–0.12), which was attributed to the rather small database. Furthermore, a so-called optimal brain surgeon (OBS) method was used to prune the ANN through the identification and removal of irrelevant nodes (weight elimination). The architectures as well as exemplary 3D profiles for predicting the wear rate in dependency of the SCF and the TiO$_2$ content before and after pruning are illustrated in Figure 4. Apparently, both cases matched adequately with the experimental data. Besides higher computational efficiency, the pruned network featured superior prediction accuracy in some areas of the parameter space. Finally, optimal compositions with higher SCF and lower TiO$_2$ concentrations around 10–15% as well as 3–5%, respectively, could be derived with considerably reduced experimental efforts, which corresponded well to the observations from Jiang et al. [46]. Gyurova and Friedrich [47] evaluated the influence of the data set size on the prediction capabilities of trained ANNs. Utilizing a newly measured database consisting of 124 independent pin-on-disk dry sliding wear tests on PPS matrix composites, the mean relative errors were reduced from above 0.72 to below 0.55 (specific wear rate) and from above 0.11 to beneath 0.10 (COF) compared to previous studies [45,48]. Later, the approach was further enhanced by Busse and Schlarb [49] using the same data, most notably by utilizing a LM training algorithm with mean squared error regularization as performance function, which significantly improved the computational efficiency and, in particular, the accuracy. Independently of the inputs, the wear rate prediction quality was found to be six times higher compared to the comparative studies [45,47].

Zhu et al. [50] also emphasized the crucial role of data set size and reported better agreement of experimental data with the prediction of the friction coefficient than with the volumetric wear losses when applying an ANN to carbon fiber and TiO$_2$ reinforced PTFE. 12 Different compositions were therefore investigated in block-on-disk dry sliding tests under varying sliding velocities and normal loads. A network trained by gradient search and consisting of three hidden layers (15, 10, and 5 neurons) and tan-sigmoid transfer functions between the input and the hidden layers as well as pure linear transfer functions to the output layer was found to deliver the least mean square errors. Li et al. [51] applied a Monte Carlo-based ANN to predict the tribological behavior of PTFE resin with aramid pulp, potassium titanate whisker (PTW), mica, copper (Cu) as well as silicon dioxide (SiO$_2$) for ultrasonic motors and compared the performance to a back propagation ANN. The database, an orthogonal table by variation of the composition, was generated from experiments conducted in triplicate on a quasi-static test rig where the specimens were fixed on a dynamic rotor and slid against a phosphor bronze stator at constant speed and load. In combination with a grey relational analysis, it was shown that especially mica and SiO$_2$ exerted significant roles for friction and wear improvements. The Monte Carlo-based ANN was particularly suitable for predictions with more limited amount of data due to repeated random sampling and the utilization of combinations of different transfer functions (sigmoid, polynomial, tanh, and gauss functions). The authors reported that, in the context of the variation and volatility of the underlying data, the Monte Carlo ANN

performed better than the conventional back propagation ANN with root mean squared errors of 0.97 (specific wear rate) and 0.007 (COF) compared to 2.08 and 0.019.

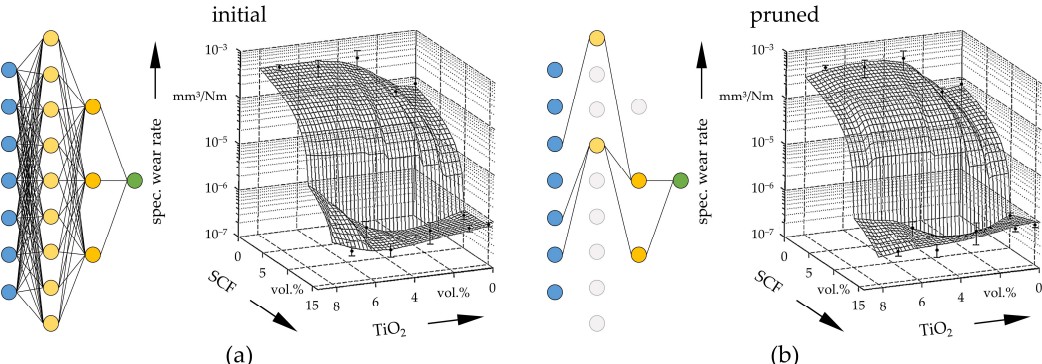

**Figure 4.** ANN architecture as well as 3D profiles for the specific wear rate in dependency of the SCF and the TiO$_2$ concentration without (**a**) and with (**b**) pruning. Redrawn from [45] with permission (Elsevier).

Kurt and Oduncuoglu [52] utilized 125 data sets extracted from established literature sources to study the effects of normal load and sliding speed in dry sliding experiments as well as the type and weight fraction of various reinforcements in ultrahigh molecular weight PE (UHMWPE) composites by a feed forward back propagation ANN. This involved zinc oxide (ZnO), zeolite, carbon nanotubes (CNT), carbon fibers (CF), graphene oxide (GO), and wollastonite additives, leading to a total number of 11 inputs, whereas the volumetric wear loss was considered as target/output value. In a trial-and-error search, an ANN with a single-hidden layer consisting of 12 neurons and logistic sigmoid transfer functions trained by a LM algorithm was selected. With $R^2$ values for training and testing above 0.8 as well as mean absolute errors not exceeding 4.1%, it was thus shown that sliding speed and load determined the wear losses more significantly than the particle types and fractions. Recently, Vinoth and Datta [53] also used 153 experimental data sets from literature to predict mechanical properties of UHMWPE composites with multi-walled carbon nanotubes (MWCNT) and graphene reinforcements in dependency of seven input variables comprising composite composition, particle size, and mechanical bulk properties. A feed forward ANN with scaled conjugate gradient back propagation, hyperbolic tangent transfer functions and 3 (for Young's modulus) or 5 (for the ultimate tensile strength) hidden layers were utilized, achieving correlation coefficients for the outputs of 0.93 and 0.97, respectively. Subsequently, a multi-objective (pareto) optimization of the input variables was performed with a non-dominated sorting genetic algorithm. On this basis, samples (pins) of UHMWPE composites with MWCNT and graphene filler ratios considered as optimal were fabricated accordingly and characterized mechanically as well as in tribological tests under dry sliding conditions against cobalt chromium alloy disks. It was actually possible to demonstrate improved properties compared to references and, in particular, excellent wear behavior due to the formation of wear-protecting transfer films on the counter-body.

### 3.1.3. Metal Matrix Composites

Some successful studies using ML and AI can also be found for composites with soft metals as matrix [54], for example aluminum, copper or zinc and their alloys [55–59]. As such, Stojanović et al. [60] investigated the friction and wear behavior of aluminum hybrid composites with Al-Si alloy matrix and 10 wt.% silicon carbide (SiC) as well as 0, 1, and 3 wt.% graphite. The data sets were generated in lubricated block-on-disk tribometer tests at three sliding speeds, the normal loads and at constant sliding distance with the application of Taguchi's robust orthogonal array design method (27 data points). This was reported to be a simple and efficient methodology. Besides performing ANOVA factor variance analysis and the fitting of a linear regression model, a feed forward back propagation

ANN was developed. Therefore, 70% of the data were used for training, 15% for testing and 15% for validation. The model was trained by LM optimization and consisted of two hidden layers of 20 and 30 neurons, respectively, as well as logarithmic sigmoid and pure linear transfer functions. The values predicted by the ANN provided sufficient agreement with the experiments and were more precise than those provided by the statistical methods used. Similarly, Thankachan et al. [61] compared the performance of a feed forward back propagation ANN with statistical regression analysis when investigating the wear behavior of hybrid copper composites with aluminum nitride and boron nitride particles in dry-running pin-on-disk tribometer tests at different volumetric fractions, loads, sliding speeds and sliding distances by applying Taguchi's orthogonal array. The ANN featured the 4 inputs, one hidden layer with 7 neurons, the specific wear rate as output and was trained by the LM function to optimize the mean absolute error. Thus, the neural network reached higher accuracy than the reference regression model.

Gangwar and Pathak [62] introduced a novel improved bat algorithm (IBA) to train an ANN for predicting the wear behavior of marble dust reinforced zinc-aluminum (Zn-Al) alloy by optimizing the weights, biases and neurons as well as finding minimum mean squared errors, see Figure 5a). The main advantage of the IBA compared to other training algorithms (e.g., back propagation, genetic algorithms or particle swarm optimization) was in the flexibility and stable training through the introduction of a new velocity, position search equation and sugeno inertia weights. This overcame local optima stagnation and enhanced the convergence speed. The evaluation of the specific wear rate was based on data from pin-on-disk experiments with varied filler content, normal load, sliding velocity and distance, as well as ambient temperature (5 levels each) by means of a Taguchi orthogonal array (25 data sets). Thereby, an ANN with 7 neurons in the single-hidden layer was found to be optimal, with a mean squared error of 0.26 and an average prediction accuracy of 97%. Exemplary 3D plots of the wear rate as a function of the variation parameters are shown in Figure 5b). Obtained results and the suggested IBA-ANN approach can thus help to save resources when searching for beneficial stress or material combinations with limited experimental database.

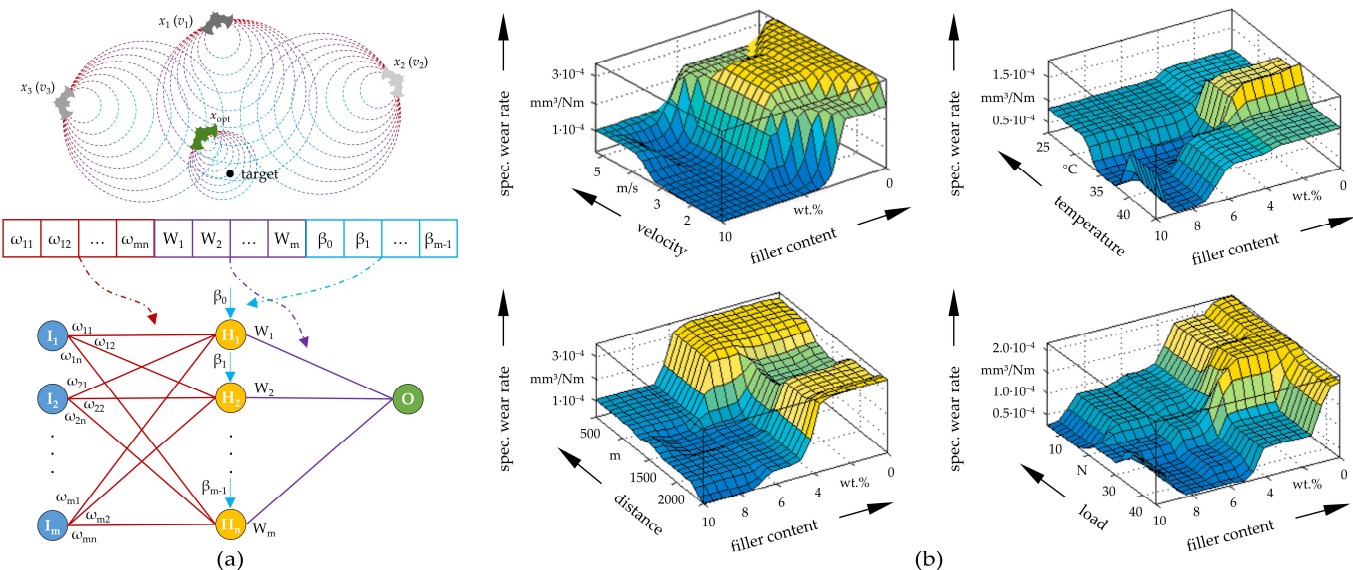

**Figure 5.** Schematic representation of encoding bat individuals to train the ANN (**a**) as well as 3D response surfaces of influencing factors on the specific wear rate (**b**). Redrawn from [62] with permission (Elsevier).

Very recently, Hasan et al. [63,64] compared five different ML techniques when predicting the friction and wear behavior of aluminum base alloys and graphite composites: ANN, kNN, SVM, gradient boosting machine (GBM), and RF. The 852 data sets were obtained from experimental studies in literature. It was shown that basically all ML approaches

were able to adequately describe the tribological behavior from material and tribological test data. Thereby, RF outperformed the other algorithms in predicting the wear behavior, while GBM and KNN had the highest accuracy for the friction behavior for the base alloy and the composite, respectively. This underlines that the right choice of the ML approach is highly dependent on the respective problem formulation.

The works in the area of composite materials are summarized in Table 1 according to the subject, the database, the inputs and outputs, and the ML approach.

### 3.2. Drive Technology

In the field of drive technology, there are several areas of application for using ML for rolling and sliding bearings, seals, brakes, and clutches, which are involved in systems for motion generation and power transmission.

### 3.2.1. Rolling Bearings

Rolling bearings are among the most important machine elements, locally transmitting large forces via several rolling contacts. The bearing components are also exposed to complex dynamics and friction occurs in numerous contacts influencing the operation. Bearing failures can be of very different nature. Mostly, they are longer lasting processes between first occurrence of damage and fatal failure. However, damage to rolling bearing components can be quickly observed in the operating behavior of machines and systems, for example in the form of increasing friction, heat, vibration, and noise. Therefore, one possible application for ML is condition monitoring and damage detection [65,66]. Most published work was related to vibration theory rather than tribology [67,68], which is why only some representative examples shall be introduced.

**Table 1.** Overview of ML approaches successfully applied in the area of composite materials.

| Subject | Database, Number of Data Sets (If Applicable Divided in Train/Test/Validation) | Inputs | Outputs | ML Approach | Prediction | Ref. |
|---|---|---|---|---|---|---|
| SGF and BFS reinforced epoxy | experimental (pin-on-disk) Taguchi DoE, 16 | BFS content, sliding velocity, normal load, sliding distance | spec. wear rate | back propagation ANN (4:7:1) | Errors < 6.9% | [39] |
| unidirectional short castor oil fiber reinforced epoxy | experimental (pin-on-disk) full factorial DoE, 36 (60%/20%/20%) | fiber length, normal load, sliding distance | wear, temperature, COF | various ANNs, best results for back propagation ANN (3:9:3 & 3:9:12:9:3) | averaged total errors < 5% | [40] |
| treated betelnut fiber reinforced polyester | experimental (block-on-disk), 492 | fiber orientation, normal load, sliding distance | COF | ANN (3:30:20:1) | SSE < 1% | [41] |
| glass fiber reinforced polyester | experimental (disk-on-flat), 7389 | fiber orientation, rotational speed, normal load, test duration | COF | ANN (4:40:1) | SSE < 15% | [42] |
| SCF, graphite, PTFE, and TiO$_2$ reinforced PPS | experimental (pin-on-disk), 90 (80%/20%) | matrix vol. fraction, filler, reinforcing agent and lubricant, contact pressure, sliding speed, tensile strength, compressive strength | spec. wear rate, COF | various gradient descent back propagation ANNs (7:9:3:1 for wear, 7:3:1:1 for COF) | MRE < 0.78 (wear), MRE < 0.12 (COF) | [45] |
| | experimental (pin-on-disk), 124 (80%/20%) | | | | MRE < 0.55 (wear), MRE < 0.10 (COF) | [47] |
| | | | | | MRE < 0.14 (wear), MRE < 0.03 (COF) | [49] |

Table 1. *Cont.*

| Subject | Database, Number of Data Sets (If Applicable Divided in Train/Test/Validation) | Inputs | Outputs | ML Approach | Prediction | Ref. |
|---|---|---|---|---|---|---|
| CF and TiO$_2$ reinforced PTFE | experimental (block-on-ring), 30–105 (10–98%/2–90%), best results for largest database | PTFE content, carbon fiber content, TiO$_2$ content, sliding speed, normal load, hardness, compressive strength | vol. wear loss, COF | various ANNs, best results for gradient search ANN (7:15:10:5:1) | CoD > 90% | [50] |
| aramid pulp, PTW, mica, Cu, and SiO$_2$ reinforced PTFE | experimental (rotor/stator test-rig) in orthogonal table DoE, 18 (80%/20%) | aramid pulp content, PTW content, mica content, Cu content, SiO$_2$ content | spec. wear rate, COF | back propagation ANN | RMSE < 2.08 (wear), RMSE < 0.019 (COF) | [51] |
| | | | | Monte Carlo-based ANN | RMSE < 0.97 (wear), RMSE < 0.007 (COF) | |
| ZnO, zeolite, CNT, CF, GO, and wollastonite reinforced UHMWPE | experiments from literature, 125 | UHMWPE content, ZnO content, Zeolite content, CNT content, CF content, GO content, wollastonite content, normal load, sliding speed | vol. wear loss | back propagation ANN (11:12:1) | R$^2$ > 0.8, mean total error < 4.1% | [52] |
| MWCNT and graphene reinforced UHMWPE | experiments from literature, 153 | MWCNT fiber diameter, MWCNT fiber length, MWCNT content, graphene sheet length, graphene sheet thickness, graphene content, UHMWPE molecular weight, UHMWPE tensile strength, UHMWPE Young's modulus | Young's modulus, tensile strength | scaled conjugate gradient back propagation ANN (7:3:1 for Young's modulus and 7:5:1 for tensile strength) | R$^2$ > 0.93 (Young's modulus), R$^2$ > 0.97 (tensile strength) | [53] |
| graphite reinforced Al-Si alloy | experimental (block-on-disk) in Taguchi's orthogonal array DoE, 27 (70%/15%/15%) | graphene content, normal load, sliding speed | vol. wear rate, COF | back propagation ANN (3:20:30:2) | R$^2$ > 0.98 | [60] |
| aluminum nitride and boron nitride reinforced copper | experimental (pin-on-disk) in Taguchi's orthogonal array DoE, 27 (90%/10%) | volume fraction, normal load, sliding velocity, sliding distance | spec. wear rate | back propagation ANN (4:7:1) | errors < 3.4% | [61] |
| marble dust reinforced Zn-Al alloy | experimental (pin-on-disk) in Taguchi's orthogonal array DoE, 25 (60%/20%/20%) | filler content, normal load, sliding velocity, sliding distance, amb. temperature | spec. wear rate | IBA trained ANN (5:7:1) | MSE < 0.26, accuracy > 97% | [62] |

**Table 1.** *Cont.*

| Subject | Database, Number of Data Sets (If Applicable Divided in Train/Test/Validation) | Inputs | Outputs | ML Approach | Prediction | Ref. |
|---|---|---|---|---|---|---|
| Graphite reinforced aluminum alloy | experiments from literature, 852 | graphite content, hardness, ductility, processing procedure, heat treatment, SiC content, yield strength, tensile strength, normal load, sliding velocity, sliding distance, | vol. wear rate, COF | back propagation ANN (11:10:10:10:2) | MSE < 0.003 wear) RMSE < 0.06 (wear) $R^2$ > 0.74 (wear) MSE < 0.004 (COF) RMSE < 0.06 (COF) $R^2$ > 0.86 (COF) | [63,64] |
| | | | | kNN | MSE < 0.002 wear) RMSE < 0.04 (wear) $R^2$ > 0.85 (wear) MSE < 0.007 (COF) RMSE < 0.08 (COF) $R^2$ > 0.76 (COF) | |
| | | | | RF | MSE < 0.001 wear) RMSE < 0.04 (wear) $R^2$ > 0.88 (wear) MSE < 0.004 (COF) RMSE < 0.06 (COF) $R^2$ > 0.86 (COF) | |
| | | | | SVM | MSE < 0.006 (COF) RMSE < 0.08 (COF) $R^2$ > 0.76 (COF) | |
| | | | | GBM | MSE < 0.002 wear) RMSE < 0.04 (wear) $R^2$ > 0.86 (wear) MSE < 0.003 (COF) RMSE < 0.05 (COF) $R^2$ > 0.89 (COF) | |

As such, Subrahmanyam and Sujatha [69] investigated the suitability of two different ANNs, namely multilayered feed forward neural network trained with supervised error back propagation (EBP) technique and an unsupervised adaptive resonance theory-2 (ART2) based neural network, for the diagnosis of local defects in deep groove ball bearings. The input vector consisted of eight parameters that were used to describe the vibration signal and the output was a condition rating for the bearing (good/bad) and, if the condition was classified bad, the defect was pinpointed. The authors concluded from their work that the performance of the ANN with EBP was excellent for recognizing ball bearing states. They reported that defective bearings were distinguished from good ones with 100% confidence, while the ANN had a success rate of over 95% in diagnosing localized defects. The results of the ANN with ART2 were ambivalent: The learning process was about 100 times faster than that of the ANN with EBP and defective bearings also were distinguished from good ones with 100% reliability. Yet, the estimation of localized defects was not satisfactory. Furthermore, Kanai et al. [70] presented a condition monitoring method for ball bearings using both, model-based estimation (MBE) and ANN, to guess the vibration velocity and the defect frequency of the rotor-bearing-system. The authors based their study on a three-layered feed forward neural network trained with EBP, where the input vector consisted of 5 parameters (speed, load, defect volume, radial clearance, number of balls) obtained from rig tests on a self-aligning deep groove ball bearing. According to the authors, the ANN shows satisfactory results compared to MBE and experimental tests.

Apart from condition monitoring, ML approaches have recently been utilized for designing rolling bearing components. Schwarz et al. [71] used different ML methods to

classify possible cage motion modes of rolling bearings and to predict application-related undesired cage instabilities, see Figure 6. The data set was generated from sophisticated rolling bearing dynamics simulations, which were confirmed by means of experimental investigations on a test rig. Based on the simulations, the authors determined metrics that, in combination, reliably characterize the state of the cage condensed in three classes "stable", "unstable" and "circling". They used these metrics to classify cage motion using quadratic discriminant analysis (QDA). QDA is a method of multivariate statistics to separate different classes on the basis of characteristics [16]. It is interesting to note that we could not discover this method in any other article within our literature survey. To predict the class of cage motion, Schwarz et al. applied decision trees as weak learners within an ensemble classification model based on AdaBoostM1 [72] to achieve good results. Furthermore, Wirsching et al. [73] aimed at tailoring the roller face/rib contact in tapered roller bearings. Geometric parameters were sampled by a Latin hypercube sampling (LHS) and the tribological behavior was predicted by means of elastohydrodynamic lubrication (EHL) contact simulations. Key target variables such as pressure, lubricant gap and friction were approximated by a so-called metamodel of optimal prognosis (MOP) [74] and optimization was carried out using an evolutionary algorithm (EA). The MOP fully automatically filtered non-significant variables and various approaches (polynomial regression, moving least squares, isotropic or anisotropic kriging) were trained to derive the most suitable approximation. The applied ML approach provided very good prediction for most geometries and target values, which was reflected in the high prediction coefficients (CoP) in most cases above 90% and the low errors in mostly below 2% of the optimized pairing between the prediction and verification calculations.

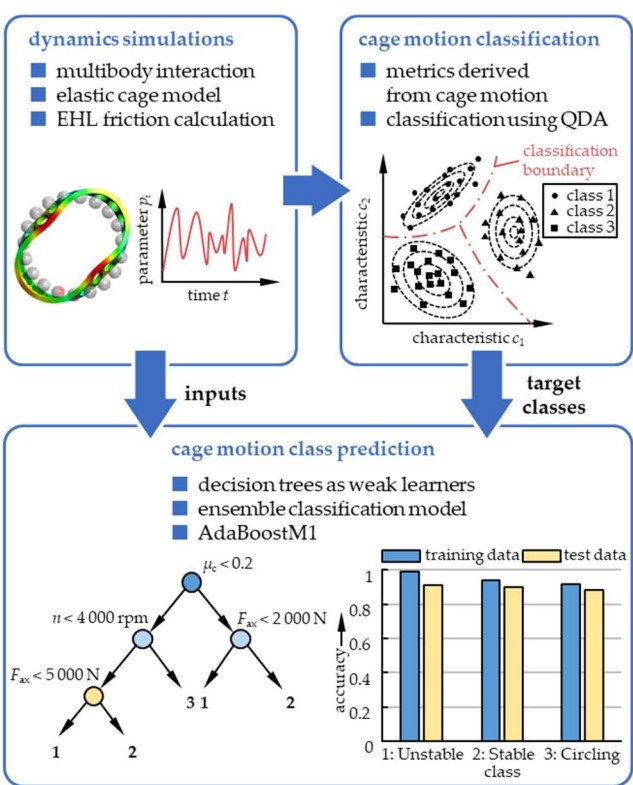

**Figure 6.** Global scheme for classifying and predicting rolling bearing cage motion modes based on dynamics simulations and ML following [71].

### 3.2.2. Sliding Bearings

Since the operating behavior of sliding bearings is highly non-linear and depending on numerous parameters, ML methods have been utilized for the analysis and synthesis of the tribosystem. Canbulut et al. [75] analyzed the frictional losses of a hydrostatic slipper

bearing using an ANN fed by experimental test data. Input parameters were the average roughness of the rubbing surfaces, relative velocity, supply pressure, hydrostatic pocket ratio, and capillary tube diameter. Three-layered feed forward neural networks containing 10 neurons in the hidden layer trained with EBP were to be found as suitable. The predictive performance of the ANN was evaluated using six operation cases for the bearing, where an exact match of the ANN predictions with the experimental results was reported. Further, using ANNs, Ünlü et al. [76] analyzed the friction and wear behavior of a radial journal bearing (bronze CuSn10/steel SAE 1050 pairing) under dry and lubricated conditions. The ANN with EBP technique was featured a 3:5:5:3 multilayer architecture for the dry case and 3:4:4:3 for the lubricated case. The input vector was described by time, applied load and rotational speed and the outputs were coefficient of friction, journal and bearing weight loss. Input data were collected from previously published experiments. The ANN predictions show high agreement to the experimental data and the authors stated that such ANNs can effectively reduce the number of future experiments. Furthermore, Moder et al. [77] showed that supervised ML algorithms can be used to predict the lubrication regime of hydrodynamic radial journal bearings based on given torques. Therefore, the torque time series were first analyzed using Fast Fourier Transformation (FFT) and manually assigned to lubrication regimes. Two ML algorithms were used for the classification task: Logistic regression and deep neural networks. Based on their results, the authors concluded that even shallow neural networks as well as logistic regressions are able to reach high accuracy for the given problem. It was indicated that data scaling was essential, while feature scaling, which is often applied in data analysis, was not suitable for the FFT classification. Prost et al. [78] investigated the feasibility of classifying the operating condition (running-in, steady, pre-critical, critical) of a translationally oscillating self-lubricating journal bearing using an ensemble learning algorithm. To this end, the authors applied a semi-supervised random forest classifier (RFC), which was based on the aggregation of a large number of independent decision trees. The RFC was trained with high-resolution force signals from experiments and showed a very high classification accuracy in validation experiments. The authors pointed out, that labeling the data is essential and requires expert knowledge. As this step is very tedious and time-consuming, they suggested a semi-automated process based on principal components analysis and k-means clustering algorithms. Francisco et al. [79] studied how far ML can be used to optimize connecting rod big-end bearings. They combined sophisticated finite element (FE) simulations with a nondominated sorting genetic algorithm, which allowed them to minimize the frictional losses and functioning severity of the bearing by optimizing 10 parameters. The authors concluded that metamodels based on previous simulations and including all relevant parameters allow the optimization of a tribological system in a very time and resource saving way.

### 3.2.3. Seals

Seals play an important role in mechanical drive technology as they separate lubricants or operating fluids and the environment of the drive train from each other. Contact seals frequently affect the friction behavior in the whole drive train, and they are exposed to wear. Increasing requirements demand more precise descriptions of the tribological behavior of contact seals in design phases as well as condition monitoring [80,81]. Logozzo and Valigi [82] suggested ANNs as an alternative for analytical models to predict friction instabilities and critical angular speeds of face seals during shaft decelerations. The authors studied different feed forward neural networks with 2:$x$:1 architecture ($x$ = 6, 8, 10, 12, 15, 16), trained with supervised EBP technique. Thereby, 10 neurons in the hidden layer showed the best training convergence. Input data were collected from experimentally validated tribo-dynamics simulations based on a lumped parameter model with 2 degrees of freedom. The input vector of the ANNs consisted of two parameters (axial and torsional stiffness). The authors pointed out that unlike deterministic models, the ANNs were not able to explain the phenomena of frictional instability but provided a smart

way to define parameters in the design phase for the avoidance of frictional instabilities. Yin et al. [83] used a SVM regression to monitor the status of a gas face seal based on acoustic emissions (AE). Input data as well as validation data were collected from rig tests. To generate the representative vectors with satisfactory experimental agreement, the AE power concentration in several key bands within certain durations were used.

### 3.2.4. Brakes and Clutches

Brakes and clutches are safety-relevant components and have to work reliably even under extreme conditions. They are usually integrated in closed loop control systems, which makes it necessary to describe and optimize the braking and coupling behavior, involving complex squealing and wear phenomena. Accordingly, ML approaches have been applied in this area as well [84,85]. For example, Aleksendrić et al. [86,87] applied an ANN to model the speed-dependent cold performance of brakes They considered 18 material composition parameters, five manufacturing parameters and three operating condition parameters as inputs. Friction data was collected from test rig experiments. Since it is not known a priori which model provides the best prediction quality, the authors investigated 18 different architectures with five different learning algorithms (LM, Bayesian regulation, resilient back propagation, scaled conjugate gradient and gradient decent). The best prediction results were provided by a 26:8:4:1 double-hidden-layer architecture trained by a Bayesian regulation algorithm. The authors stated that their ANN has shown sufficient flexibility to generalize the influences of unknown types of friction material on their cold performance. The methodology was later extended to predict materials recovery performance [88] and brake wear [89] by the same authors. Basically, the procedure was similar to the work described above and the best prediction results were attained from a single hidden layer ANN (25:5:1) trained by a Bayesian regulation algorithm. Timur and Aydin [90] investigated whether the friction coefficient of brakes can be predicted by means of ML based upon experimental training data (1050 points). Comparing different regression methods (linear, least median squared linear, Gaussian processes, pace, simple linear, isotonic, SVM) and 10-fold cross-validation, they noted that all algorithms showed a correlation coefficient larger than 0.99 and a root mean squared error below 0.01. However, isotonic regression allowed the fastest model building.

The prediction of friction coefficient for automobile brake as well as clutch materials against steel using ML algorithms was also addressed by Senatore et al. [91], who showed how to obtain a comprehensive view on the influence of the main sliding parameters. Based upon experimental data from pin-on-disk tests with varying sliding speed, acceleration and contact pressure (200 data sets), the authors trained two different supervised feed-forward double-hidden-layer EPB ANNs with a 3:6:3:1 architecture for braking and 3:6:7:1 for the clutch material, respectively. The authors concluded that ANNs have confirmed suitability for valid prediction of friction coefficients, with utility being enhanced by significance as well as sensitivity analysis of input parameters. A possible application could be in more accurate friction maps for electronic control purpose. However, the authors also discussed the limitations of the approach, in particular pointing out extrapolation errors. Comparable findings were obtained by Grzegorzek and Scieszka [92], who used a similar methodology (in this case feed-forward EPB ANN with 6:12:1 architecture) to investigate the friction behavior of industrial emergency brakes from 408 data sets. The authors self-described their work as being at a preliminary stage, yet they were able to demonstrate the performance of ANN against various models of multiply regression analysis.

The works in the area of drive technology are summarized in Table 2 according to the subject, the database, the inputs and outputs, and the ML approach.

Table 2. Overview of ML approaches successfully applied in the area of drive technology.

| Subject | Database, Number of Data Sets (If Applicable Divided in Train/Test/Validation) | Inputs | Outputs | ML Approach | Prediction | Ref. |
|---|---|---|---|---|---|---|
| groove ball bearing defect diagnosis | experimental (bearing test-rig), 108 (90%/10%) | peak value of amplitude, average of top five peak values of amplitude, peak value of auto-correlation function, standard deviation, kurtosis | bearing state | EBP ANN | success rate > 95% | [69] |
| | | | | ART2 ANN | success rate = 100% | |
| ball bearing condition monitoring | experimental (bearing test-rig), 145 (75%/25%) | speed, load, defect volume, radial clearance, number of balls | vibration velocity | back propagation ANN (5:12:1) | errors < 14% | [70] |
| cage motion mode classification in rolling bearings | numerical (dynamics simulation) LHS, 4000 | cage mass, cage bending stiffness, pocket clearance, guidance clearance, bearing type, COF, axial force, radial force, bending moment, rotational speed | CDI | QDA and DT | accuracy > 91% | [71] |
| TRB roller/face rib contact geometry design | numerical (EHL simulation) in LHS, 370 (70%/30%) | roller face radius, eccentricity, rib radius | max. pressure, min. film height, COF | MOP | CoP > 90%, errors < 2% | [73] |
| frictional power losses of hydrostatic slipper bearings | experimental (hydrostatic slipper test-rig) | average roughness, relative velocity, supply pressure, hydrostatic pocket ratio, capillary tube diameter | frictional power loss | back propagation ANN | errors < 1.9% | [75] |
| dry and lubricated journal bearing behavior | experimental (journal bearing test-rig), 4 | time, load, rotational speed | COF, bearing weight loss, journal weight loss | EBP ANN (3:5:5:3 for dry and 3:4:4:3 for lubricated case) | mean errors < 4% (dry), mean errors < 5.3% (lubricated), | [76] |
| journal bearing lubrication regime prediction | experimental (journal bearing test-rig), 888 (80%/20%) | frictional torque | lubrication regime | FFT+ back propagation ANN (1:256:128:64: 32:16:8:1) | accuracy > 99% | [77] |
| journal bearing operating condition classification | experimental (journal bearing test-rig), 9 (75%/25%) | time, lateral force | operating state | RFC (DT) | accuracy > 94% | [78] |

**Table 2.** *Cont.*

| Subject | Database, Number of Data Sets (If Applicable Divided in Train/Test/Validation) | Inputs | Outputs | ML Approach | Prediction | Ref. |
|---|---|---|---|---|---|---|
| connecting rod big-end bearing design | numerical (elastic HL simulation) in CCF DoE, 9 | oil viscosity at ref. temperature, oil viscosity at ref. pressure, oil thermo-viscosity coefficient, oil piezo-viscosity coefficient, oil piezo-viscosity index, oil supply pressure, lemon shape, shell bore relief depth, shell bore relief length, barrel shape, radial clearance | pressure times velocity product, power loss | nondominated sorting genetic algorithm | $R^2 > 0.99$ | [79] |
| face seal friction instability prediction | numerical (dynamics simulation), 40 (90%/10%) | axial stiffness torsional stiffness | critical speed | various ANNs, best results for EBP ANN (2:10:1) | $R^2 > 0.97$ | [82] |
| disk brake performance | experimental (inertial dynamometer), 275 (70%/10%/20%) | applied pressure, initial speed, number of braking events, phenolic resin, iron oxide, barites, calcium carbonate, brass chips, aramid, mineral fiber, vermiculite, steel fiber, glass fiber, brass powder, copper powder, graphite, friction dust, molybdenum disulphide, aluminum oxide, silica, magnesium oxide, spec. molding pressure, molding temperature, molding time, heat treatment temperature, heat treatment time | brake factor | various ANNs, best results for Bayesian ANN (26:8:4:1) | sufficient (not quantified) | [86, 87] |
| brake materials | experimental (inertial dynamometer), 408 (34%/33%/33%) | sliding speed, contact pressure, temperature, binder resin, premix masterbatch, residuum | COF | EPB ANN (6:12:1) | errors < 4% | [92] |
| clutch materials | experimental (pin-on-disk), 200 (50%/25%/25%) | sliding speed, sliding acceleration, contact pressure | COF | EPB ANN (3:6:3:1) / EPB ANN (3:6:7:1) | sufficient within the data range (not quantified) | [91] |

*3.3. Manufacturing*

ML approaches were also employed in the area of manufacturing technology, for example, for process monitoring or in quality control/image recognition [6]. There are also some studies related to tribology, particularly regarding friction stir welding [93–95], but

also for forming or machining [96]. Sathiya et al. [97] modeled the relationship between friction welding process parameters as heating pressure, heating time, upsetting pressure as well as upsetting time and output parameters (tensile strength and metal loss) when joining similar stainless steel by means of a back propagation ANN with 9 neurons in the single-hidden layers. The database (14 data points) was generated from corresponding experiments. Subsequently, different optimization strategizes based upon the ANN's prediction were compared: Genetic algorithm, simulated annealing algorithm, and particle swarm optimization. Among them, the genetic algorithm was reported to be most suitable and good agreement was found between the prediction of tensile strength and metal loss for optimized process parameters with respective validation experiments. Similarly, Tansel et al. [98] and Atharifar [99] applied and confirmed the suitability of ANNs for optimizing friction stir welding processes. However, the latter further introduced an optimization of the back propagation ANNs using a genetic algorithm (genetically optimized neural network systems) to maximize the prediction quality. Anand et al. [100] compared the performance of an ANN (4:9:2) with a response surface methodology approach (quadratic polynomial models) when optimizing friction welding with respect to tensile strength and burn-off length. The data (30 data sets) were generated with experiments within a five-level, four variable centrale composite DoE (CCD). It was observed that the ANN featured higher accuracy by a factor of two compared to the response surface. In turn, Dewan et al. [101] compared back propagation neural networks with adaptive neuro-fuzzy interference systems (ANFIS) [102] when predicting tensile properties in dependency of spindle speed, plunge force and welding speed from a rather small database (73 data points). Here, 1200 different ANFIS models were developed with varying number and type of membership functions as well as input combinations. It was reported the optimized ANFIS provided lower prediction errors than the ANN.

In addition to process optimization, ML approaches have also been used for monitoring friction stir welding. Baraka et al. [103] made use of process signals (traverse and downward tool force) to predict the weld quality. This was based upon frequency analysis by FFT, and an interval type 2 radial base function (RBF) neural network trained by an adaptive error propagation algorithm that effectively provided continuous feedback to the operator with an accuracy above 80%. Das et al. [104] also used real-time process signals (torque) for internal defect identification in friction welding. The experimentally obtained signals were analyzed by discrete wavelet transformation, statistical features (dispersion, asymmetry, excess) as well as general regression models and ML methods, namely SVM and back propagation ANN (3:5:1, log-sigmoid transfer functions) trained by the gradient descent method to predict tensile strength. It was reported the prediction performance of the SVM (0.5% error) was superior to regression (13.6%) and the ANN (3.1%).

Regarding other manufacturing processes, Fereshteh-Saniee et al. [105] trained a feedforward back propagation ANN with 21 neurons in the single hidden-layer (tan-sigmoid transfer function) from over 700 FE simulations to determine material flow and friction factors in one-step ring forming. Thereby, obtained load curves showed good agreement with experimental validation tests, featuring an accuracy of 99% and 97% for grease lubricated and dry conditions, respectively. The difference was traced back to higher variations of friction for unlubricated forming. Furthermore, Bustillo et al. [106] attempted to predict surface roughness and mass loss during turning, grinding, or electric discharge machining based upon surface isotropy levels and different ML approaches: Artificial regression trees, multilayer perceptions (MLP), RBF networks, and random forest. The most accurate approach for predicting the loss of mass was found to be RBF, while the MLP most precisely predicted the arithmetic mean roughness. However, the model parameters of both approaches had to be tuned very carefully and even small changes led to a substantial increase of errors. In contrast, satisfactory accuracy without any tuning stage could be obtained using the random forest ensembles. It was also reported that the prediction quality was comparatively sound even outside the training record as well as for smaller data sets.

The works in the area of manufacturing technology are summarized in Table 3 according to the subject, the database, the inputs and outputs, and the ML approach.

**Table 3.** Overview of ML approaches successfully applied in the area of manufacturing technology.

| Subject | Database, Number of Data Sets (If Applicable Divided in Train/Test/Validation) | Inputs | Outputs | ML Approach | Prediction | Ref. |
|---|---|---|---|---|---|---|
| friction stir welding process optimization | experimental (friction stir welding), 14 | heating pressure, heating time, upsetting pressure, upsetting time | tensile strength, metal loss | back propagation ANN (4:9:2) | MSE < 0,01% | [97] |
| | experimental (friction stir welding), 30 | | | | RMSE < 0.98 (tensile strength), RMSE < 0.05 (tensile strength), | [100] |
| | experimental (friction stir welding), 73 (60%/20%/20%) | rotational speed, welding speed, plunge force, empirical force index | tensile strength | various ANNs, best results for back propagation ANN (3:5:1) | mean absolute error < 7.7% | [101] |
| | | | | ANFIS | mean absolute error < 10.1% | |
| friction stir welding process monitoring | experimental (friction stir welding), 25 (80%/20%) | rotational speed, welding speed | weld threshold for downward force, weld threshold for traverse force | RBF trained ANN | accuracy > 80% | [103] |
| | experimental (friction stir welding), 64 (60%/25%/15%) | rotational speed, welding speed, shoulder diameter | tensile strength | SVM | error < 0.5% | [104] |
| | | | | back propagation ANN | error < 3% | |
| ring forming | numerical (FE simulation), 700 | polynomial regression factors to fit load-displacement curves | strain hardening exponent, strength coefficient, COF | ANN (8:21:3:3) | accuracy > 97% | [105] |

### 3.4. Surface Engineering

Approaches to enhance the tribological behavior of components by modifying their surfaces can be subsumed under the term surface engineering [107]. This involves adjusting the surface topography with and without compositional changes through as well as the application of coatings. Examples include, among others, tailoring the roughness and/or statistically distributed or discrete micro-textures, carburizing, nitriding, anodizing, electroplating, weld hardfacing, thermal spraying, chemical, or physical vapor deposition (CVD, PVD) [107]. Some studies have also applied ML approaches to better understand or design the surface modifications.

#### 3.4.1. Coatings

Cetinel [108] used a single-hidden layer feed forward ANN to predict the COF and wear loss of thermally sprayed aluminum titanium oxide ($Al_2O_3$-$TiO_2$) coatings. The database was created by reciprocal pin-on-block tribometer tests under dry as well as acid conditions different loads. In the ANN, the test conditions were the inputs and—after

trial-and-error testing of different configurations—the hidden layer consisted of 80 neurons. Furthermore, the ANN provided 63 outputs in the form of the COF and linear wear progress at different times of the experiments. Thus, the tribological behavior over the test period could be mapped very well. Sahraoui et al. [109] analyzed the friction and wear behavior of high-velocity oxy-fuel (HVOF) sprayed Cr-C-Ni-Cr and WC-Co coatings as well as electroplated hard chromium by means of a feed forward ANN. The database consisted of 180 training and 180 test data sets from dry-running pin-on-disk tribometer tests of the coated test specimens against brass disks at various normal loads and sliding speeds. An ANN with sigmoid transfer functions as well as two hidden layers (6 and 4 neurons) was found to be suitable for predicting the COF within variabilities between 5.8% and 10.8%. The main advantage of the model in this study was that the friction coefficients could be predicted comparatively well for a range of parameters up to 7 times larger than those contained in the training data. Upadhyay and Kumaraswamidhas [110,111] applied a back propagation ANN to optimize multilayer nitride coatings on tool steel deposited by unbalanced reactive PVD magnetron sputtering. The input parameters comprised bias voltage and gas flow rate as well as time, velocity and load within pin-on-disk sliding tests. The data was split into 70% training, 15% validation, and 15% test data. Training was based on the LM function and the most favorable ANN consisted of 20 neurons in the hidden layer. Thus, the wear rate as well as the COF could be predicted within errors of less than 10%.

3.4.2. Surface Texturing

Otero et al. [112] attempted to optimize surface micro-textures fabricated by photolithography and chemical etching processes in order to reduce the COF of EHL contacts by means of an ANN. The data was obtained from tests on a mini-traction machine (steel ball-on-micro-textured copper disk) at various loads, total speeds and slip conditions. The ANN consisted of 7 inputs (average velocity, SRR, load, minor and major axis dimensions, depth and texturing density), 20 neurons in the hidden layer, and the COF as output. Thus, load case-dependent ranges for beneficial texture parameters could be derived. Additionally, referring to tests on samples with pores or micro-textures on a lubricated mini traction machine at different test conditions, Boidi et al. [113] applied an RBF to predict the wear behavior of sintered components. The database included 1704 experimental sets with different sum velocities and slip, as well as geometric or statistical characteristics of the dimples, grooves and pores, respectively. A Hardy multiquadric RBF was found to provide an excellent fit with an overall correlation of 0.93, especially with regard to the standard deviations of the tribological experiments. Mo et al. [114] utilized statistical methods as well as a back propagation ANN with 60 neurons in the single-hidden layer to investigate the role of micro-texture shape deviations and dimensional uncertainties on the tribological performance. The database was founded on physical modeling approaches in the form of simulations of parallel, hydrodynamically lubricated (HL) contacts and randomly split into 70% training and 30% validation data. The trained ANN was able to predict the relationships between geometric micro-texture parameters (e.g., dimple diameter, depth, area density etc.) and the frictional force as well as the load carrying capacity with an accuracy of 99.7% and 97.5%, respectively. Thus, the influences of statistical deviations (e.g., roundness errors, standard deviations of the dimensional parameters, etc.) could be estimated and optimal, robust optima could be retrieved by means of a genetic algorithm. Similarly, Marian et al. [115,116] utilized a MOP [74] to model the influence of micro-textures in EHL contacts as well as an EA to optimize the micro-texture geometry and distribution. Based upon a LHS (70 data sets) and contact simulations, the contact pressure, lubricant film height, and frictional force were predicted with CoPs larger than 82%, allowing subsequent optimization with an EA. Zambrano et al. [117] used reduced order modeling (ROM) to predict and optimize the frictional behavior of surface textures in dynamic rubber applications under different operating conditions. It is noteworthy that this was based on a limited number of experimental measurements and the ROM was

fed with microscope-based texture measurements. In this sense, besides nominal texture parameters, the real geometries as well as their deviations and uncertainties have been evaluated with good accuracy.

The works in the area of surface engineering are summarized in Table 4 according to the subject, the database, the inputs and outputs, and the ML approach.

**Table 4.** Overview of ML approaches successfully applied in the area of surface engineering.

| Subject | Database, Number of Data Sets (If Applicable Divided in Train/Test/Validation) | Inputs | Outputs | ML Approach | Prediction | Ref. |
|---|---|---|---|---|---|---|
| thermally sprayed $Al_2O_3$-$TiO_2$ coatings | experimental (pin-on-disk), 8 | load, environment (dry or acid) | linear wear, COF at different time steps | back propagation ANN (2:80:63) | sufficient (not quantified) | [108] |
| HVOF sprayed Cr-C-Ni-Cr and WC-Co coatings and electroplated hard chromium | experimental (pin-on-disk), 360 (50%/50%) | material type, normal load, sliding velocity, sliding distance | COF | back propagation ANN (4:6:4:1) | errors < 11% | [109] |
| multilayer nitride PVD coatings | experimental (pin-on-disk), 246 (70%/15%/15%) | time, normal load, sliding velocity, lap, bias voltage, gas flow rate | spec. wear rate, COF | back propagation ANN (6:5:5:2) | errors < 1% | [110, 111] |
| surface texture design for EHL contacts | experimental (mini traction machine), 2000 (90%/5%/5%) | average velocity, slide-to-roll ratio, normal load, minor axis, major axis, texture depth, texture density | COF | various ANNs, best results for back propagation ANN (7:20:1) | MSE < 0,1%, $R^2$ > 0.99 | [112] |
| | experimental (mini traction machine), 1704 | entrainment speed, slide-to-roll ratio, surface feature ball, surface feature disk | COF | Hardy multiquadric RBF | $R^2$ > 0.935 | [113] |
| | numerical (EHL simulation) in LHS, 70 (70%/30%) | texture diameter, texture depth texture distance | max. pressure, min. film height, COF | MOP | CoP > 83% | [115, 116] |
| surface texture design for HL contacts | numerical (HL simulation) | dimple diameter, depth, area density, and various statistical deviations | COF, load carrying capacity | various ANNs, best results for back propagation ANN (41:20:2) | accuracy > 99.7% (COF), accuracy > 97.5% (load carrying capacity) | [114] |

### 3.5. Lubricants

ML/AI approaches have also been used in the development and formulation of lubricants [118] and their additives [119] intended for the use in tribological systems. As such, Durak et al. [120] analyzed the effects of PTFE-based additives in mineral oil onto the frictional behavior of hydrodynamic journal bearings (252 data sets) by the aid of a feed forward back propagation ANN. An architecture with three inputs as studied in respective experiments (load, velocity, additive concentration), two hidden layers of 5 and 3 neurons, and the COF as output resulted in an accuracy of 98%. Therefore, optimal concentrations depending on the load case could be identified with rather little

experimental effort. Humelnicu et al. [121] applied an ANN to investigate the tribological behavior of vegetable oil-diesel fuel mixtures. The data were generated in pin-on-disk tests at constant conditions, whereas the concentration of rapeseed and sunflower oil was varied and the averaged COF values of five repetitions of each combination was used for further processing. The neural network was trained with a back propagation algorithm and tangential transfer functions and the architecture considered as most suitable with relative deviations between 0.2% and 2.3% was built of three hidden layers with 2, 6, and 9 neurons, respectively. Bhaumik et al. [122,123] also applied a multi-hidden layer feed forward ANN to design lubricant formulations with vegetable oil blends (coconut, castor and palm oil) and various friction modifiers (MWCNT and graphene) based upon 80 data sets obtained from four-ball-tests as well as 120 data sets from pin-on-disk tests as reported in various literature. The respective material and test conditions were also included as influencing factors. For building the ANN, hyperbolic tangent transfer functions and a scaled conjugate gradient back propagation algorithm were used. Good prediction quality was thus achieved for the 11 and 13 inputs in the four-ball- and pin-on-disk tests, respectively, with accuracies over 92%. In addition to the influences of the lubricant and material properties, significant differences were also revealed due to the test setup. In an optimization based on the ANN using a genetic algorithm, it was also possible to derive ideal lubricant formulations, the suitability of which was actually demonstrated by subsequent preparation and corresponding experimental validation. Lately, Mujtaba et al. [124] utilized a Cuckoo search algorithm to optimize an extreme learning machine (ELM) and a response surface methodology (RSM) in predicting the tribological behavior of biodiesel from palm-sesame oil in dependency of ultrasound-assisted transesterification process variables. Based on a Box-Behnken experimental design, the biodiesel yield was predicted, whereby the ELM featured a better performance than RSM, and optimized. In tribological experiments on a four-ball-tester, improved friction and wear behavior compared to reference lubricants was also demonstrated with the derived blend.

In addition to these more macro-tribological approaches, some studies can also be found that tend to target even smaller scales [125]. For example, Sattari Baboukani et al. [126] employed a Bayesian modeling and transfer learning approach to predict maximum energy barriers of the potential surface energy, which corresponds to intrinsic friction, of various 2D materials from the graphene and the transition metal dichalcogenide (TMDC) families when sliding against a similar material with the aim of application as lubricant additives. The input variables for the model in the form of different descriptors (structural, electronic, thermal, electron-phonon coupling, mechanical and chemical effects) were extracted from density function theory (DFT) and molecular dynamics (MD) simulation studies in literature. The applied Bayesian model accommodated the sparse and noisy data set and estimated the maximum energy barrier as target variable as well as its uncertainty and potentially missing data. The predictions were validated against MD simulations, whereas excellent agreement with mean squared errors mostly below 0.25 were found. Thus, the application of the ML approach not only allowed for the prediction estimation of the applicability for tribological purposes of ten previously underexplored 2D materials, but also initiated discussion on novel empirical correlations and physical mechanisms.

The works in the area of lubricant formulation are summarized in Table 5 according to the subject, the database, the inputs and outputs, and the ML approach.

Table 5. Overview of ML approaches successfully applied in the area of lubricant formulation.

| Subject | Database, Number of Data Sets (If Applicable Divided in Train/Test/Validation) | Inputs | Outputs | ML Approach | Prediction | Ref. |
|---|---|---|---|---|---|---|
| PTFE-based additives in mineral oil | experimental (journal bearing test-rig), 252 (80%/20%) | load, velocity, additive concentration | COF | back propagation ANN (3:5:3:1) | accuracy > 98% | [120] |
| vegetable oil-diesel fuel mixtures | experimental (pin-on-disk), 135 | sunflower concentration, rapeseed concentration | COF | back propagation ANN (2:2:6:9:1) | RMSE < 0,1% | [121] |
| lubricant formulations with vegetable oil blends and friction modifiers (MWCNT, graphene) | literature (pin-on-disk, four-ball-tests), 200 | speed, normal load, temperature, ball/pin/disk hardness, coconut oil content, castor oil content, palm oil content, MWCNT content, MWCNT size, graphene content, graphene dimensions | COF | scaled conjugate gradient back propagation ANN | accuracy > 92% | [122, 123] |
| biodiesel formulation | experimental (transesterification), 30 | time, catalyst concentration, methanol-to-oil ratio, duty cycle | biodiesel yield | RSM | $R^2$ > 0.994, MSE < 0.023, RMSE < 0.151 | [124] |
| | | | | Cuckoo ELM | $R^2$ > 0.996, MSE < 0.024, RMSE < 0.117 | |
| lubricant additives | literature and numerical (DFT and MD simulation) | lattice constant, c/a ratio, bond angle, interlayer space, M-X length, X-X length, M-radii, hexagonal width, in-plane stiffness, cohesive energy, binding energy, bandgap energy, thermal conductivity, average mass | maximum energy barrier | Bayesian model | MSE < 0.25 | [126] |

*3.6. Others/General*

Apart from aforementioned areas, a wide variety of studies can be found in fields which also related to tribology, but that were not assigned to the traditional core and are therefore not included in more detail in this review. The tribology of driven piles in clay [127], plate tectonics and earthquakes [128], or motion control [129,130] can be mentioned as examples. Nevertheless, some selected research shall be presented that did not necessarily fit into one of the upper categories but had a rather general scope. As such, already in 2002, Ao et al. [131] introduced an ANN to predict the evolution of surface topography during the wear process. The proposed approach utilized surface measurements at a finite number of time intervals during tribological experiments in a conformal block-on-ring configuration. The back-propagation ANN with sigmoid transfer functions was trained with the LM algorithm and statistical surface parameters (RMS roughness, skewness, kurtosis, and autocorrelation). Together with initial surface parameters, the corresponding 3D topography in worn conditions could be estimated by surface synthesis. Thereby, good prediction quality could be achieved, especially if

the autocorrelation function did not experience stronger changes. So, this was not just about predicting and optimizing target variables but was rather already a step towards semi-physical modeling. Thereby, the usage of ML/AI in the field of tribology may not be limited to forward approaches, which predict the tribological behavior based on some input data sets in the context of an experimental design. Accordingly, Haviez et al. [132] later developed a modified ANN model, which was used to solve actual physical equations describing the phenomena of fretting wear. Interestingly, this eliminated the necessity for iterative learning, e.g., by back propagation, or other regularization techniques. Thus, the fretting wear damage could be predicted with higher efficiency and accuracy than by a conventional back propagation ANN trained with experimental data, highlighting the ability of generalization albeit the rather low level of complexity. Similarly, Argatov and Chai [133] suggested an ANN-based modeling framework for analyzing the dry sliding wear during running-in from pin-on-disk tribometer tests. The authors attempted to derive the true wear coefficient instead of the specific wear rate at given conditions, contact pressures and sliding velocities. This was based upon the integral and differential forms of the Archard's wear equation as well as single-hidden layer ANN with sigmoid transfer functions. They applied their approach to various data from the literature ranging from cermet coatings, zirconia reinforced aluminum hybrid composites to nickel–chromium alloys and reported good efficiency and agreement. Very recently, Almqvist [134] derived a physics informed neural network (PINN) to solve the initial and boundary value problems described by linear ordinary differential equations and to solve the second order Reynolds differential equation. Thereby, comparable results to analytical solutions were obtained. The advantage of the present approach is not in accuracy or efficiency, but in the fact that it is a mesh-free method that is not data-driven. The author hypothesized that this concept could be generalized in the future and lead to a more accurate and efficient solution of related but nonlinear problems than the currently available routines.

Finally, two papers shall be highlighted that addressed other approaches than ANNs and/or other scales as well. Bucholz et al. [135] used a dataset from dry sliding pin-on-disk tests with different ceramic pairings having different intrinsic properties and inorganic minerals to develop a predictive model. The latter was generated by the recursive partitioning method, resulting in a graphical expression of the classification of observations according to similarities determined by variable importance in projection and the error some of squares. The obtained regression tree as illustrated in Figure 7a) demonstrated a satisfactory coefficient of determination above 0.89 when comparing prediction and experiment (Figure 7b). Finally, Perčic et al. [136] recently trained various ML/AI approaches to predict the nanoscale friction of alumina ($Al_2O_3$), titanium dioxide ($TiO_2$), molybdenum disulphide (MoS2), and aluminum (Al) thin films in dependency of several process parameters, including normal forces, sliding velocities, and temperature. The data were acquired by lateral force microscopy (LFM) within a centroidal Voronoi tessellation (CVT) design of experiments, whereas 2/3 of the data were generally used for training and 1/3 for validation. The study employed MLP ANN, random DT and RF, support vector regression (SVR), age-layered population structure (ALPS), grammatical evolution (GE), and symbolic regression multi-gene programming (SRMG). The suitability for predicting the frictional force for these approaches was further evaluated with respect to the mean absolute error, the root mean squared error and the coefficient of determination. Thereby, the SRMG model showed the best performance with prediction accuracies (determination coefficient) between 72% and 91%, depending on the sample type. This allowed to derive simple functional descriptions of the nanoscale friction for studied variable process parameters.

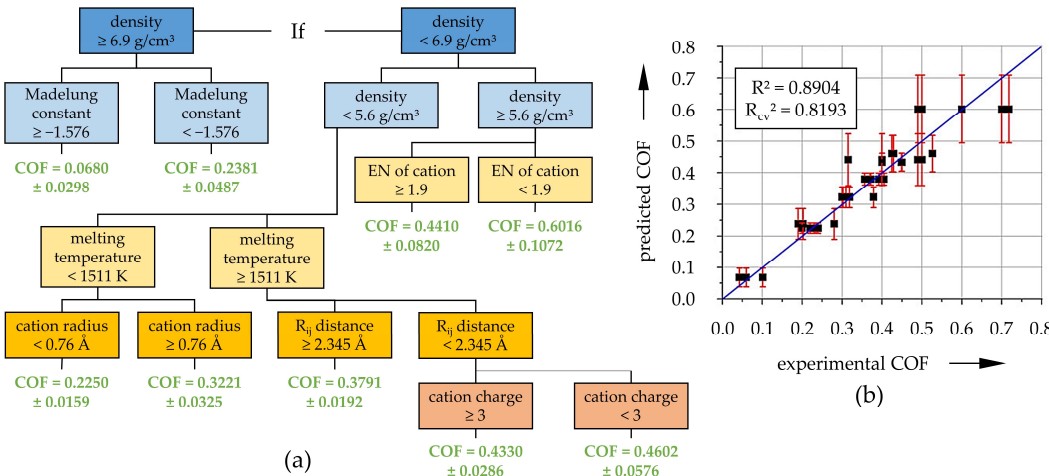

**Figure 7.** Dendrogram for the COF estimation from recursive partitioning (**a**) and comparison between experimental and predicted values (**b**). Redrawn from [135] with permission (Springer).

## 4. Summary and Concluding Remarks

Tribology naturally involves multiple interacting features and processes, where machine learning and artificial intelligence approaches are feasible to support sorting through the complexity of patterns and identifying trends on a much larger scale than the human brain is capable of. Computers are able to fit thousands of properties, which enables for a much wider search of the available solution space and allows quantitative fits to a broad range of properties. Predictions do not have to be limited to averaged or global values/outputs but could also cover locally and timely resolved evolutions and bridge the gap between different scales. Therefore, ML and AI might change the landscape of what is possible going beyond the mere understanding of mechanisms towards designing novel and/or potentially smart tribological systems. As is also evident from the quantified survey, ML has hence already been employed in many fields of tribology, from composite materials and drive technology to manufacturing, surface engineering, and lubricants. The intent of ML might not necessarily be to create conclusive predictive models but can be seen as complementary tool to efficiently achieve optimum designs for problems, which elude other physically motivated mathematical and numerical formulations. We assume that, besides the availability of larger amounts of experimental data, this is the reason for the comparatively large number of investigations on composite materials.

The challenge is that a ML approach does not necessarily guide towards the specific problem solution and the selection as well as optimization of a qualified algorithm is of decisive importance. Accordingly, there is a wide variety of approaches that have already been successfully applied to answer tribological research questions. A summary is provided in Table 6, which is—together with Tables 1–5—intended to support researchers in identifying initial selections.

**Table 6.** Overview of ML approaches successfully applied in various areas of tribology.

| ML Approach | Composite Materials | | | Drive Technology | | | | Manufacturing | | | Surface Engineering | | Lubricants | Others |
| --- | --- | --- | --- | --- | --- | --- | --- | --- | --- | --- | --- | --- | --- | --- |
| | Thermoset Matrix | Thermoplastic Matrix | Metal Matrix | Rolling Bearings | Sliding Bearings | Seals | Brakes And Clutches | Friction Stir Welding | Forming | Machining | Coating | Texturing | | |
| ANN | [39–42] | [43–53] | [60–64] | [69,70] | [75–77] | [82] | [86–89, 91,92] | [97–101,103,104] | [105] | [106] | [108–111] | [112,114] | [120–123] | [131–134, 136] |
| ANFIS | | | | | | | | [101] | | | | | | |
| Bayesian | | | | | | | | | | | | | [126] | |
| DT | | | [63,64] | [60] | | | | | | [106] | | | | [135, 136] |
| KNN | | | [63,64] | | | | | | | | | | | |
| MOP | | | | [73] | | | | | | | | [115, 116] | | |
| QDA | | | | [71] | | | | | | | | | | |
| RF | | | [63,64] | | [78] | | | | | [106] | | | | [136] |
| RBF | | | | | | | | | | [106] | | [113] | | |
| SVM | | | [63,64] | | | [83] | [90] | [104] | | | | | | [136] |

Apparently, a large share of the research discussed in this article (roughly three quarters) was based on ANNs. However, even still, there are manifold possibilities concerning architecture, training algorithms, or transfer functions. Other ML approaches are still less commonly used for tribological issues but are justifiably coming more into focus and can be more effective for some problems. The reproducibility and comparability of the prediction quality from the various approaches and studies is frequently hampered by the sometimes ambiguous underlying database and the lack of information on the implementation of ML approaches withing publications as well as the use of different error/accuracy measures. Most of the works also comprised forward ML models, which were developed to predict the tribological behavior as output based on various input parameters such as material or test conditions. In principle, however, inverse models to characterize the materials and surfaces [54] or physics-informed ML approaches [134] can also be applied. With a closer assessment of the intentions and objectives of the studies, as well as the overrepresentation of ANNs, one might get the impression that ML is in many cases being used to serve its own ends. The added value compared to physical modeling or statistical evaluation based on more classical regressions is not always evident. A few studies, however, manage to extract real insights and thus additional knowledge from a large and broad database. The comprehensive works in the field of composite materials from Kurt and Oduncuoglu [52], Vinoth and Datta [53], and Hasan et al. [63,64] utilizing literature-extracted databases may be highlighted here and can serve as excellent examples. The current showstopper is still the availability of sufficient and comparable datasets as well as the handling of uncertainties regarding test conditions and deviations. In this respect, we would like to encourage authors to also publish the underlying databases and the corresponding models in appendices or data repositories. Moreover, there is great potential to automatize and optimize the data acquisition and processing, which is presently still very manual in the field of tribology, in order to unfold the knowledge already available in institutes, enterprises or in the literature by means of machine learning.

**Author Contributions:** Conceptualization, M.M.; methodology, M.M. and S.T.; formal analysis and investigation, M.M. and S.T.; data curation, M.M.; visualization, M.M.; writing—original draft preparation, review and editing, M.M. and S.T. Both authors have read and agreed to the published version of the manuscript.

**Funding:** This research received no external funding.

**Institutional Review Board Statement:** Not applicable.

**Informed Consent Statement:** Not applicable.

**Data Availability Statement:** The data on the quantitative evaluation are available upon request from the corresponding author.

**Acknowledgments:** The authors kindly acknowledge the continuous support of Friedrich-Alexander-University Erlangen-Nuremberg (FAU) and University of Bayreuth, Germany.

**Conflicts of Interest:** The authors declare no conflict of interest.

## Abbreviations

| | |
|---|---|
| AE | acoustic emission |
| AI | artificial intelligence |
| ALPS | age-layered population structure |
| ANFIS | adaptive neuro-fuzzy interference system |
| ANN | artificial neural network |
| ART | adaptive resonance theory |
| BFS | blast furnaces slag |
| CCD | centrale composite design |
| CDI | cage dynamics indicator |
| CF | carbon fiber |
| CMC | ceramic matrix composite |
| CNT | carbon nanotube |
| CoD | coefficient of determination |
| COF | coefficient of friction |
| CoP | coefficient of prognosis |
| CVT | centroidal voronoi tessellation |
| DFT | density function theory |
| DoE | design of experiments |
| DT | decision tree |
| EA | evolutionary algorithm |
| EBP | error back propagation |
| EHL | elastohydrodynamic lubrication |
| ELM | extreme learning machine |
| FE | finite element |
| FFT | fast fourier transformation |
| GBM | gradient boosting machine |
| GE | grammatical evolution |
| GO | graphene oxide |
| HL | hydrodynamic lubrication |
| HVOF | high-velocity oxy-fuel |
| IBA | improved bat algorithm |
| kNN | k-nearest neighbor |
| LFM | lateral force microscopy |
| LHS | latin hypercube sampling |
| LM | levenberg-marquardt |
| MBE | model-based estimation |
| MD | molecular dynamics |
| ML | machine learning |
| MLP | multilayer perception |

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
