# Peer review of "Current Trends and Applications of Machine Learning in Tribology—A Review"

_lubricants, doi:10.3390/lubricants9090086_

Round 1

Reviewer 1 Report

The present manuscript is well-organized and reflects current trends of ML and AI in lubricants. The manuscript can be accepted in present form. 

Author Response

Dear editors, dear reviewers,

we would like to thank you for your feedback and the opportunity to resubmit a revised version of our manuscript entitled "Current Trends and Applications of Machine Learning in Tribology – A Review ".

We would also like to take this opportunity to express our thanks to the learned reviewers for the positive feedback and helpful comments. Based on the instructions provided by the reviewers, we uploaded the file of the revised manuscript with changes highlighted in red. Furthermore, we have responded specifically to each suggestion below.

We hope the revised version will be received favorably and are looking forward to hearing from you in the near future.

Sincerely yours

Max Marian and Stephan Tremmel

Point-by-point responses

Reviewer #1:

The present manuscript is well-organized and reflects current trends of ML and AI in lubricants. The manuscript can be accepted in present form.

Response:

The authors are grateful and would like to thank the reviewer for his/her positive feedback.

Reviewer #2:

The article raises an interesting topic about the current approach of Machine learning and Artificial Intelligence in the field of tribology. The paper is well written, well organized,  it will be interesting to a wide range of readers and has decent scientific values to be published in the Lubricants journal. I propose to accept the article in its present form.

Response:

We thank the reviewer for his/her detailed evaluation of the manuscript and the positive feedback.

Reviewer #3:

I appreciate the authors for putting together this review manuscript discussing the emerging filed of "triboinformatics". It was very needed for the further progress of this field. Here are few comments to further improve the paper:

Response:

The authors are particularly grateful for the positive feedback as well as the detailed and specific comments. We have addressed mentioned points below (changes in the manuscript are highlighted in red) and believe that the manuscript was substantially improved after suggested edits.

In addition, the reviewer noted that moderate improvements in English are required. Even though the language and readability were rated as very good (5/5 stars) by the other two reviewers, we have carefully reviewed the article again with regard to the linguistic expressions.

Comment #1:

Line 186. From here onward, please add the number of data points used for training ML algorithms in each study

Response:

Thank you for that suggestion, we have revised the manuscript accordingly and added the number of data sets in the text and/or the table (see comment #2).

Comment #2:

In the "result" section, it would be beneficial if you can draw tables summarizing the various studies discussed in each section. In the tables, you can include what parameters were used as inputs/outputs for the ML models, the type of ML algorithm and its details, the number of train/test/validation data points, details of the materials and tribological test condition, etc.

Response:

We are grateful for this prescient comment. We have revised the manuscript accordingly and believe that the manuscript’s readability and accessibility was substantially improved after suggested edits. To even further enhance this, we have also added a list of abbreviations at the end of the manuscript.

Comment #3:

In most studies covered by this manuscript, forward ML models were developed in order to predict the tribological properties based on various inputs such as materials or test conditions. However, this is only one way to apply ML/AI for materials design and processing (please refer to the article below). Have the authors come across studies where the inverse models were used for the optimization of process parameters, or classification methods were developed to characterize microstructure of the worn materials, etc.?

"Kordijazi, A., Zhao, T., Zhang, J., Alrfou, K., & Rohatgi, P. (2021). A Review of Application of Machine Learning in Design, Synthesis, and Characterization of Metal Matrix Composites: Current Status and Emerging Applications. JOM, 1-15."

Response:

Thank you for this comment and making us aware of the interesting review paper, which was now referenced as well. Indeed, there are some publications that dealt with mentioned aspects. We have now highlighted this distinction into forward and inverse as well as physics-informed ML in the manuscript (directly when discussing the papers and in the concluding remarks) and believe that the interested reader can benefit from this clarification.

Reviewer 2 Report

The article raises an interesting topic about the current approach of Machine learning and Artificial Intelligence in the field of tribology. The paper is well written, well organized,  it will be interesting to a wide range of readers, and has decent scientific values to be published in the Lubricants journal. I propose to accept the article in its present form. 

Author Response

(The authors gave the same response as above.)

Reviewer 3 Report

I appreciate the authors for putting together this review manuscript discussing the emerging filed of "triboinformatics". It was very needed for the further progress of this field. Here are few comments to further improve the paper:

-Line 186. From here onward, please add the number of data points used for training ML algorithms in each study

-In the "result" section, it would be beneficial if you can draw tables summarizing the various studies discussed in each section. In the tables, you can include what parameters were used as inputs/outputs for the ML models, the type of ML algorithm and its details, the number of train/test/validation data points, details of the materials and tribological test condition, etc.

-In most studies covered by this manuscript, forward ML models were developed in order to predict the tribological properties based on various inputs such as materials or test conditions. However, this is only one way to apply ML/AI for materials design and processing (please refer to the article below). Have the authors come across studies where the inverse models were used for the optimization of process parameters, or classification methods were developed to characterize microstructure of the worn materials, etc.?

"Kordijazi, A., Zhao, T., Zhang, J., Alrfou, K., & Rohatgi, P. (2021). A Review of Application of Machine Learning in Design, Synthesis, and Characterization of Metal Matrix Composites: Current Status and Emerging Applications. JOM, 1-15."

Author Response

(The authors gave the same response as above.)

Round 2

Reviewer 3 Report

NA